# AgentCF: Collaborative Learning with Autonomous Language Agents for Recommender Systems

## ABSTRACT

Recently, there has been an emergence of employing LLM-powered agents as believable human proxies, based on their remarkable decision-making capability. However, existing studies mainly focus on simulating human dialogue. Human non-verbal behaviors, such as item clicking in recommender systems, although implicitly exhibiting user preferences and could enhance the modeling of users, have not been deeply explored. The main reason lies in the gap between language modeling and behavior modeling, as well as the incomprehension of LLMs about user-item relations.

To address this issue, we propose **AgentCF** for simulating user-item interactions in recommender systems through agent-based collaborative filtering. We creatively consider not only users but also items as agents, and develop a collaborative learning approach that optimizes both kinds of agents together. Specifically, at each time step, we first prompt the user and item agents to interact autonomously. Then, based on the disparities between the agents' decisions and real-world interaction records, user and item agents are prompted to reflect on and adjust the misleading simulations collaboratively, thereby modeling their two-sided relations. The optimized agents can also propagate their preferences to other agents in subsequent interactions, implicitly capturing the collaborative filtering idea. Overall, the optimized agents exhibit diverse interaction behaviors within our framework, including user-item, user-user, item-item, and collective interactions. The results show that these agents can demonstrate personalized behaviors akin to those of real-world individuals, sparking the development of next-generation user behavior simulation. Code is available at: https://anonymous.4open.science/r/AgentCF-WWW/.

## CCS CONCEPTS

• **Information systems** → **Recommender systems**.

## KEYWORDS

Agents, Large Language Models, Collaborative Learning

**ACM Reference Format:**
Anonymous Author(s). 2022. AgentCF: Collaborative Learning with Autonomous Language Agents for Recommender Systems. In *Proceedings of Make sure to enter the correct conference title from your rights confirmation email (Conference acronym 'XX)*. ACM, New York, NY, USA, 17 pages. https://doi.org/XXXXXXX.XXXXXXX

## 1 INTRODUCTION

With the recent advancements in large language models (LLMs), LLM-powered agents demonstrate impressive capabilities in autonomous interaction and decision making [23, 30, 43, 58]. By scaling their numbers, these agents exhibit the emergence of human-like behaviors at both the individual and population levels [12, 37]. This observation highlights the potential of employing LLM-powered agents to simulate human social behaviors, such as daily lives in Smallville [37] and software development [38].

Typically, existing studies primarily focus on simulating human dialogue using semantic knowledge of LLMs. However, in addition to dialogue, real-world human behaviors also involve non-verbal aspects like user-item interactions in recommender systems, which implicitly reflect user preferences and have the potential to facilitate personalized user modeling. To simulate such behaviors, some work verbalizes these interaction records into natural language text to prompt LLMs [43, 46]. Nevertheless, we argue that this method struggles to capture the underlying behavioral patterns of these interactions, due to the gap between universal language modeling and personalized behavior modeling. For example, shoppers who buy diapers on Friday also tend to buy beer [35]. This behavioral pattern can be effectively captured by collaborative filtering models [16, 39], but confuse LLMs as the two items are semantically unrelated. Therefore, it is crucial to develop methods to better characterize human behaviors in LLM-powered simulations.

Focused on this research topic, in this work, we take recommender systems, a common web application, as the testbed to study the simulation of user-item interactions with LLM-power agents. Typically, in addition to directly prompting LLMs based on user historical interactions [9, 44], there are also some studies that employ agents to perform recommendations, by introducing specialized tools and planning strategies [19, 48]. Especially, RecAgent [46] develops an LLM-based simulator, regarding users as agents. However, these studies mainly focus on characterizing user-side behaviors using universal LLMs, neglecting the effect of item-side modeling in this two-sided interaction process. In recommender systems, users possess personalized preferences for items, while items have potential adopters. Modeling the two-sided relations between users and items is crucial for enhancing personalized recommendations [16].

Considering these issues, our solution is inspired by the *collaborative learning* idea from previous recommendation models (*e.g.,* BPR [39] and NCF [16]). In these methods, users and items are modeled as equally important parts and the recommender is essentially a preference function that measures user-item affinity based on the two sides. During the optimization process, the parameters related to users and items are collaboratively optimized, making the recommender better fit interaction records. Therefore, to facilitate the comprehension of user-item relations by LLM-powered agents, our idea is to simulate both *user agents* and *item agents*, and mimic the optimization process of traditional recommenders.

The merits of such a method are twofold. Firstly, it can model the user-item interaction by autonomous interaction between user and item agents, instead of verbalizing the interaction as plain text [46]. Secondly, it enables mutual optimization of user and item agents to capture their two-sided relations, rather than simulating user behaviors individually. We are also aware of several studies, such as Reflexion [41] and Self-Refine [32], optimize general-purpose agents for better decision-making in downstream tasks. However, these methods are task-focused or user-focused, optimizing agents through their own exploration, which is essentially *self-learning* rather than *collaborative learning* emphasized in our approach.

To this end, we propose the agent-based collaborative filtering approach, named **AgentCF**. Unlike previous studies that mainly focus on user simulation, our approach considers not only users but also items as agents. Both kinds of agents are equipped with memory modules, maintaining the simulated preferences and tastes of potential adopters. This involves their intrinsic features as well as acquired behavioral information. As the key of our approach, we collaboratively optimize the user agents and item agents, leveraging the remarkable decision-making and reflection abilities of LLMs. At each step, we prompt user and item agents to autonomously interact, thereby exploring whether these simulated agents can make consistent decisions with real-world interaction records. Then, based on the feedback obtained from these interactions, we design a collaborative reflection mechanism that enables user agents and item agents to reflect on and adjust their memory in a mutual manner. In this way, the simulated preferences of user agents and item agents mutually aggregate, and can be propagated to other agents in subsequent interactions. Thus, it implicitly models the collaborative filtering idea through interactions.

We evaluate our proposed approach through extensive experiments on real-world datasets. Results show that our approach effectively simulates user-item interactions, achieving promising performance on recommendation tasks compared to several classical recommendation models and LLM-based recommenders. Moreover, within our framework, user and item agents engage in free communication, creating various types of interactions (user-user, item-item, and collective), wherein they exhibit personalized behaviors similar to those of real-world individuals. The results also spark the development of next-generation user behavior simulation. The main contributions of this work are as follows:

• We are the first to consider items as agents, and develop a user-item interaction simulation approach for recommender systems, emphasizing the modeling of two-sided interaction relations.

• We collaboratively optimize the user and item agents, design a collaborative reflection mechanism, and employ it to achieve mutual update of user and item memory.

• Extensive experiments demonstrate the effectiveness of our approach in simulating personalized interactions.

## 2 METHODOLOGY

In this section, we present the proposed agent-based collaborative filtering approach, named **AgentCF**. Our approach enables user agents and item agents to collaboratively learn from user-item interactions in recommender systems. The overall framework of our proposed AgentCF is depicted in Figure 1.

### 2.1 Preliminaries

To ease the understanding of our approach, we first describe the traditional recommendation setting, and then introduce our task setting and the overview of our approach.

**Traditional Recommendation Setting**. In a recommender system, there exists a set of users $\mathcal{U} = \{u\}$, a set of items $\mathcal{I} = \{i\}$, and a set of their interaction records (often grouped by users in chronological order), denoted by $\mathcal{D} = \{\langle u, i \rangle\}$. To conduct recommendation, a preference function $f(u, i)$ is built to capture the preference degree of user $u$ over item $i$, where an interacted item (*i.e.,* positive item, denoted by $i^+$) would receive a higher score than an item that the user does not interact with (*i.e.,* negative item, denoted by $i^-$). For example, BPR [39] builds a personalized pairwise ranking model that compares the preference over two candidate items, and NCF [16] employs neural networks to fit user-item interaction relationship. Subsequently, the learning of preference function $f(u, i)$ can be converted into a standard gradient-based function fitting problem based on training data $\mathcal{D} = \{\langle u, i \rangle\}$ in machine learning [3].

**Our Task Setting**. In this work, we follow the traditional recommendation setting above and incorporate LLM-powered agents into recommender systems. Different from prior studies [37, 46] only considering task or user agents, *item agents* are also included in our work and will be a crucial part of developing our approach. Notably, we slightly reuse the notations, and use $u$ and $i$ to denote a user agent and an item agent, respectively. Specifically, a user agent $u$ is expected to capture the preference of the corresponding real user, and an item agent $i$ is expected to reflect the corresponding item characteristics and potential adopters' preferences. Following such a setting, the original user-item interaction will be simulated by autonomous interaction between the user and item agents. In our work, we consider a *ranking* task that ranks a candidate list $\{c_1, \cdots, c_n\}$ for a user agent $u$ based on LLMs: $f_{LLM}(u, \{c_1, \cdots, c_n\})$. Different from traditional recommendation models, the LLM that implements $f_{LLM}(\cdot)$ will be fixed during the optimization process.

**Overview of AgentCF**. To implement the agents, the memory mechanism (*i.e.,* storing the past states, actions, and contexts [45]) and the reflection mechanism (*i.e.,* revising the agents' states or recognitions [36, 41]) have been widely explored in various task scenarios. However, existing work often employs a task-focused (*e.g.,* ReAct [55]) or user-oriented simulation approach (*e.g.,* RecAgent [46]), and the objects (*e.g.,* the items in recommender systems) involved in this process are not explicitly considered. Since the recommendation task essentially needs to model the two-sided interaction relation between users and items, we argue that item agents are important to simulate the real interaction scenario in recommender systems. Without incorporating item agents, it is difficult to explicitly model the essential idea "*like alike*" (*i.e.,* like the items that are adopted by similar users) in collaborative filtering [4]. Notably, our approach has two technical contributions in adapting autonomous language agents for recommender systems:

• *Collaborative memory-based optimization*. Instead of simply prompting LLMs with user's historical interactions [9, 31] , we aim to refine (without gradient update) both the simulated user and item agents through their mutual interactions. At each time step, these agents are prompted to conduct autonomous interactions, and then

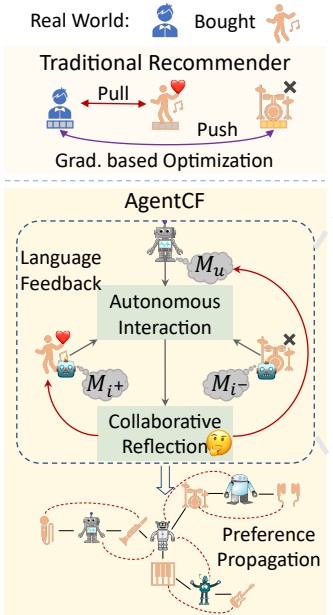

**Previous Memory**
- User Agent Memory: I adore energetic guitar-driven rock, and dance pop music…
- Pos Item Agent Memory: The CD 'Highway to Hell' is classic rock and AOR, radiating raw energy and infectious melodies that captivate fans of classic rock…
- Neg Item Agent Memory: 'The Very Best of Prince' is a Pop and Dance Pop CD, offering a collection of prince's greatest hits for an enjoyable experience…

**Autonomous Interaction**
- System Prompt: The first CD is [Memory], the second CD is [Memory]. Please select your preferred CD from these two candidates and provide an explanation.
- User Agent Response: I prefer 'The Very Best of Prince'… This CD resonates with my preference for Pop and Dance Pop CDs…

**Reflection & Memory Update**
- System Prompt: You find that you don't like the CD that you chose, indicating your preferences have changed. Please update your preferences.
- User Agent Response: I adore energetic guitar-driven rock, classic rock, and AOR. I value classic rock for its raw energy and infectious melodies. I do not like Pop…
- System Prompt: The user finds that he makes a unsuitable choice, possibly due to the misleading information in CDs' features. Please update the description.
- Pos Item Agent Response: 'Highway to Hell' is classic rock and AOR CD, exuding a raw energy and infectious melodies, ideal for energetic guitar-driven enthusiasts…

**Figure 1: The overall framework of AgentCF and a case about the optimization process of agents: (1) The user and item agents are first prompted to autonomously interact. (2) These agents adjust the misconceptions in their memory, by reflecting on the disparities between their decisions and real-world interactions. In this process, the simulated preferences of user and item agents aggregate (as indicated by the highlighted content) and can propagate to other agents in subsequent interactions.**

collaboratively reflect on the disparities between their decisions and real-world interaction records. In this way, the simulated user and item agents can accordingly adjust or update their memories, enabling them to better fit the real-world interaction behaviors.

• *Implicit preference propogation.* Different from prior studies [38, 46], a notable feature of our approach is that we update the memory of both users and item agents for each interaction record. This enables the item memory to be injected into the preferences of users who interact with it. When new interactions occur, subsequent users will be also informed about the preference of previous adopters of this item. The same process can also be applied to the user side. In this case, our approach essentially models the collaborative filtering idea by propagating preferences from user-item interactions.

## 2.2 Collaborative Agent Optimization

In this part, we present the collaborative optimization process that is developed based on both user and item agents, and discuss how it relates to the classic collaborative filtering approach.

*2.2.1 Memory Design.* To specialize the LLM-powered agents for recommender systems, different from prior studies [37, 46], we equip both user and item agents with memory modules, maintaining their intrinsic features and collaborative information.

**User Memory**. For user agents, the memory module aims to store various kinds of useful information that reflects the user preferences. Since real-world user preferences often change dynamically, we equip each user agent $u$ with short-term memory $M_u^s$ and long-term memory $M_u^l$. Especially, short-term memory is a natural language text that describes the recently updated preference of the user

agent, which can be initialized with their general preferences like "*I enjoy listening to CDs*". Furthermore, long-term memory is a pool of historical preference texts that can store the evolving process of user preference. When engaged in a new interaction, user agents can directly access their short-term memory, while retrieving relevant information from their long-term memory.

**Item Memory**. For item agents, we equip them with adjustable memory modules to record information about their own characteristics as well as the preferences of their adopters, *e.g.*, a rock CD may be ideal for energetic guitar-driven enthusiasts. However, unlike user memory, we only equip each item agent $i$ with a unified memory module $M_i$, since item information is relatively stable through time. Item memory can be initialized by their identity information, such as titles and categories, and will be continuously updated with user preferences when making new interactions. This process can largely complete the global characteristics of an item in real-world systems and enable the propagation of preference information, which is the key to collaborative learning in our approach.

*2.2.2 Autonomous Interactions for Contrastive Item Selection.* Given the initialized agents, our task is to optimize these agents and enable them to simulate real-world user-item interactions. To achieve this, we first explore the behavior alignment between these simulated agents and real-world individuals, so as to provide feedback for updating these agents. This involves testing whether the agents can make consistent decisions with real-world interaction records when selecting candidates. We can also consider this process as the "*forward computation*" of the recommender models (*e.g.*, BPR [39]) when comparing the preference over two items by a user.

Specifically, we employ the real user behavioral sequences (arranged in chronological order) as "training data". At each step of these interactions, we present a contrastive pair of both a positive item $i^+$ and a negative item $i^-$ as candidates for user agents to select from. Intuitively, agents lacking personalization mainly rely on commonsense knowledge to select popular candidates and those positioned higher in the display list for interaction. To increase the discrimination difficulty, we deliberately introduce popular bias and position bias in candidate selection: we sample a negative item with high popularity in the dataset and then place it before the positive item in the selection list. Then, the user agent $u$ is tasked with selecting an item $i^o$ from the candidates and providing explanations $y_{exp}$ for this choice, by collaboratively considering the preference from its memory and the features from the memories of candidates:

$$i^o = f_{LLM}(M_u; \ M_{i^-}; \ M_{i^+}) \tag{1}$$

$$y_{exp} = \text{Prompt}_{LLM}(i^o; \ M_u; \ M_{i^-}; \ M_{i^+}) \tag{2}$$

### 2.2.3 Collaborative Reflection and Memory Update.
The above process has enabled agents to simulate the interaction behaviors. Next, we derive the feedback signal by comparing the agents' decisions with real-world interaction data, and employ it to collaboratively optimize the involved user agent and item agents, serving as the "backward update" part of traditional recommenders. Since LLMs are fixed in this work, there is no explicit gradient-based learning process as in traditional recommender models [39]. Thus, we mainly update the associated memories of user and item agents.

Based on the agent's decision $i^o$ and explanation $y_{exp}$ generated above, we prompt both user and item agents to reflect on and adjust the misconceptions in their simulated preferences and features. We refer to this process as *collaborative reflection*, as it emphasizes the collaborative learning of the user-item relations, by performing reflection based on both user and item memories. It is different from task-specific reflection (*e.g.,* ReAct [55]) and user-oriented reflection (*e.g.,* RecAgent [46]), which construct the *self-reflection* only from the perspective of the task solver or user. Specifically, if a user agent makes the right choice that aligns with the real behavior, we inform it about the correctness and store the related interaction information in its memory. For an incorrect choice, we introduce the following collaborative reflection mechanism to revise agents' memories and behaviors accordingly, enabling user agents to simulate real interaction behaviors and the involved item agents to align with the adopter's preferences:

$$M_u^{s\prime} \leftarrow \text{Reflection}^u(i^o; \ y_{exp}; \ M_u; \ M_{i^-}; \ M_{i^+}), \tag{3}$$

$$M_i^{\prime} \leftarrow \text{Reflection}^i(i^o; \ y_{exp}; \ M_u; \ M_{i^-}; \ M_{i^+}) \tag{4}$$

$$M_u^{l\prime} \leftarrow \text{Append}(M_u^l; \ M_u^s), \tag{5}$$

where $M_u^{s\prime}$ denote the reflected short-term memory of user $u$ and $M_i^{\prime}$ denote the reflected memory of item $i$. $M_u^{l\prime}$ is the updated long-term memory of user $u$, which appends the user's previous short-term memory and can be retrieved when making a new interaction. Note that we do not modify the memory of the negative item agent, because we find that LLMs tend to over-complain the negative item agent's drawbacks, disregarding the fact that it may also be attractive for other users. From Equation (3) and (4) (a similar prompting method in Equation (2)), we can see that the reflection is actually derived based on the memories of user agent $u$, positive

item agent $i^+$, and negative item agent $i^-$. Therefore, such a reflection mechanism enables user and item agents to better understand their two-sided interaction relations by effectively discriminating between positive instances and negative instances.

At each step of interaction (*i.e.,* optimization), we iterate the selection process (Section 2.2.2) and the collaborative reflection process (Section 2.2.3) until user agents make choices consistent with those of real users or reach the maximum round of iterations. In essence, throughout the collaborative optimization process, the simulated user and item agents follow the decision-making process of real-world individuals, progressing step by step. This makes user agents become more personalized and item agents aggregate the preferences of adopters by learning from interactions, improving the simulation of user-item interactions. The implementation details and prompts are presented in Appendix B and C.

### 2.2.4 Connection with Classical Recommendation Models.
In the field of recommender systems, various models such as BPR [39] and NCF [16] are developed. In general, these models set parameters (typically hidden embeddings) to represent user preferences and item characteristics, and optimize them to fit user-item interaction records through two stages, namely *forward preference evaluation* (estimating preference score) and *backward parameter update* (performing gradient update). In this way, users or items that display similar interaction behaviors can have similar representations in parameter space, thus capturing the collaborative filtering idea.

Our approach mimics the optimization process of traditional recommenders. The user and item memories can be considered as their language-based parameters (*i.e.,* embeddings). The item selection process (Section 2.2.2) and collaborative reflection process (Section 2.2.3) correspond to the forward and backward stages in recommendation models. In our approach, explicit gradient optimization is not employed. Instead, the collaborative reflection plays a similar role of "*semantic gradient*", steering how user and item agents adjust themselves. Moreover, based on the collaborative reflection, the user and item agents mutually interact and aggregate each other's preference information, which is then propagated to new agents in subsequent interactions. Therefore, by establishing **preference propagation** from interactions of user and item agents, our approach incorporates the idea of collaborative filtering.

## 2.3 Agent Interaction Inference
After the above memory-based optimization, our approach can simulate highly personalized user agents and preference-aware item agents. In this part, we further study how to employ these simulated agents to infer the potential user-item interactions, focusing on the ranking task for a list of candidates $\{c_1, \cdots, c_n\}$.

**Basic Prompting Strategy.** Since both user and item agents are collaboratively optimized to model their two-sided relations, as the basic prompting form, we directly prompt the LLM with the agents' simulated user preferences and the candidate item features. This enables the LLM to serve as a collaborative recommender:

$$\mathcal{R}_B = f_{LLM}(M_u^s; \ \{M_{c_1}, \cdots, M_{c_n}\}) \tag{6}$$

where $M_u^s$ is the short-term memory of user agent $u$, $\{M_{c_1}, \cdots, M_{c_n}\}$ are the corresponding memories of the candidate item agents, $n$ is the number of candidates, and $\mathcal{R}$ is the ranking result.

**Table 1: Statistics of the preprocessed datasets. "Avg.c" is the average number of words in initialized item description text.**

| Datasets | #Users | #Items | #Inters. | Sparsity | Avg.c |
|---|---|---|---|---|---|
| **CDs (full)** | 93,653 | 64,032 | 1,178,439 | 99.98% | 8.04 |
| −Sparse | 100 | 704 | 800 | 98.86% | 7.76 |
| −Dense | 100 | 269 | 800 | 97.03% | 8.47 |
| **Office (full)** | 86,530 | 25,842 | 675,683 | 99.97% | 25.14 |
| −Sparse | 100 | 561 | 600 | 98.93% | 25.06 |
| −Dense | 100 | 188 | 600 | 96.81% | 25.48 |

**Advanced Prompting Strategies.** Although short-term memory describes the current preference of a user agent, retrieving their specialized preferences from long-term memory toward candidates can allow them to make more personalized inferences. In addition, when interaction records are sparse and preference propagation is limited, we can further incorporate user historical interactions into prompts, enabling LLMs to serve as sequential recommenders. These two improvement strategies can be denoted as follows:

$$\mathcal{R}_{B+R} = f_{LLM}(M_u^r;\ M_u^s;\ \{M_{c_1}, \ldots, M_{c_n}\}) \tag{7}$$

$$\mathcal{R}_{B+H} = f_{LLM}(M_u^s;\ \{M_{i_1}, \ldots, M_{i_m}\};\ \{M_{c_1}, \ldots, M_{c_n}\}) \tag{8}$$

where $M_u^r$ is the retrieved specialized preference from the long-term memory of user agent $u$ by taking the memories of candidate items as queries, and $\{M_{i_1}, \cdots, M_{i_m}\}$ are the corresponding memories of the $m$ historical interactions with items $\{i_1, \ldots, i_m\}$.

Based on the optimized user agents and item agents, our method can better simulate diverse interaction behaviors observed in the real world, such as user-item interactions, users' social behaviors, and even collective behaviors within recommender systems. Moreover, since we activate items as agents, it enables us to explore the potential feasibility of "inventing" more novel and fascinating interactions among inanimate objects. For example, item-to-item agent interaction can be useful in item cold-start scenarios by spontaneously propagating collected user preference to new items (See Section 3.4.2 for in-depth analysis experiments).

## 3 EXPERIMENTS

### 3.1 Experimental Setup

*3.1.1 Datasets.* Following previous work [17], we conduct experiments on two text-intensive subsets of Amazon review dataset [34]: "CDs and Vinyl" and "Office Products". Due to expensive API calls, we have to further sample subsets from these two datasets. Specifically, considering the effect of data sparsity on collaborative filtering recommenders, we randomly sample two subsets (one dense and one sparse) from each dataset, with each subset containing 100 users, exploring more diverse interaction scenarios. The statistics of the datasets are summarized in Table 1. Note that we use the dense datasets for further analysis experiments, as they better demonstrate the interactions among agents.

*3.1.2 Evaluation Metrics.* To evaluate the performance, we take NDCG@K as a metric, where K is set to 1, 5 and 10. Following existing studies [17, 60], we employ the *leave-one-out* strategy for evaluation. Specifically, we consider the last item in each historical

interaction sequence as the ground-truth item. By adopting the model as a ranker, we rank the target item alongside nine randomly sampled items. To further reduce randomness, we conduct three repetitions of all test instances and report the average results.

*3.1.3 Baseline Models.* We compare the proposed method with the following baseline methods:
- **BPR** [39] leverages matrix factorization to learn the representations of users and items by optimizing the BPR loss.
- **SASRec** [21] captures the sequential patterns of user historical interactions by utilizing a transformer-encoder.
- **Pop** ranks candidates based on their popularity, which is measured by the number of interactions.
- **BM25** [40] ranks candidates according to their textual similarity with user historical interactions.
- **LLMRank** [18] uses the ChatGPT as a zero-shot ranker, considering user sequential interaction histories as conditions.

For our approach, we consider three major variants: AgentCF$_B$, AgentCF$_{B+R}$, and AgentCF$_{B+H}$ (corresponding to Equation (6), (7), and (8), respectively). As our simulated user and item agents are optimized on sampled datasets, we compare our method with BPR and SASRec by training them on the sampled datasets, referred to as BPR$_{sample}$ and SASRec$_{sample}$. We also report the performance of these two models when trained on the complete dataset for reference, denoted as BPR$_{full}$ and SASRec$_{full}$. The Pop model, which relies on statistics to make recommendations, is trained on the complete datasets. Since LLMRank and BM25 are zero-shot models, training them on the complete dataset would be meaningless. Therefore, we directly evaluate their performance on the sampled datasets. Scaling agents for larger datasets is left as future work.

### 3.2 Overall Performance

We compare our approach with the baseline methods on four datasets and present the results in Table 2. We can find:

(1) Our approach outperforms other baselines in most scenarios, highlighting the effectiveness of collaborative learning for simulating personalized agents. Specifically, the performance achieved by adopting our model as a collaborative filtering recommender (AgentCF$_B$) demonstrates the efficacy of preference propagation in modeling the collaborative filtering idea. In addition, retrieving user agents' specialized preferences toward candidates (AgentCF$_{B+R}$) enables more personalized inferences. In sparse interaction cases (Office dataset), incorporating user historical interactions to enhance the prompt and enabling LLMs to serve as a sequential recommender (AgentCF$_{B+H}$) can make better performance.

(2) Our model exhibits superior or comparable performance to traditional recommendation models when trained on datasets of the same scale (sampled datasets). Furthermore, our model even achieves comparable performance to traditional recommenders trained on full datasets, when dealing with sparse scenarios. Although our model still has room for improvement in other scenarios, it is worth emphasizing that our model is trained using only approximately 0.07% of the complete dataset. These results further demonstrate the generalization capability of our approach.

(3) Existing tuning-free methods, such as Pop, BM25, and LLM-Rank, exhibit unsatisfactory performance. Specifically, although

**Table 2: Performance comparison of different models. We highlight the best and the second-best among traditional recommenders trained on sampled datasets, tuning-free models, and our approach, using bold and underlined fonts, respectively.**

| Method | $CDs_{sparse}$ | | | $CDs_{dense}$ | | | $Office_{sparse}$ | | | $Office_{dense}$ | | |
|---|---|---|---|---|---|---|---|---|---|---|---|---|
| | N@1 | N@5 | N@10 | N@1 | N@5 | N@10 | N@1 | N@5 | N@10 | N@1 | N@5 | N@10 |
| $BPR_{full}$ | 0.1900 | 0.4902 | 0.5619 | 0.3900 | 0.6784 | 0.7089 | 0.1600 | 0.3548 | 0.4983 | 0.5600 | 0.7218 | 0.7625 |
| $SASRec_{full}$ | 0.3300 | 0.5680 | 0.6381 | 0.5800 | 0.7618 | 0.7925 | 0.2500 | 0.4106 | 0.5467 | 0.4700 | 0.6226 | 0.6959 |
| $BPR_{sample}$ | 0.1300 | 0.3597 | 0.4907 | 0.1300 | 0.3485 | 0.4812 | 0.0100 | 0.2709 | 0.4118 | 0.1200 | 0.2705 | 0.4576 |
| $SASRec_{sample}$ | 0.1900 | 0.3948 | 0.5308 | 0.1300 | 0.3151 | 0.4676 | 0.0700 | 0.2775 | 0.4437 | **0.3600** | **0.5027** | **0.6137** |
| Pop | 0.1100 | 0.2802 | 0.4562 | 0.0400 | 0.1504 | 0.3743 | 0.1100 | 0.2553 | 0.4413 | 0.0700 | 0.2273 | 0.4137 |
| BM25 | 0.0800 | 0.3066 | 0.4584 | 0.0600 | 0.2624 | 0.4325 | 0.1200 | 0.2915 | 0.4693 | 0.0600 | 0.3357 | 0.4540 |
| LLMRank | 0.1367 | 0.3109 | 0.4715 | 0.1333 | 0.3689 | 0.4946 | 0.1750 | 0.3340 | 0.4728 | 0.2067 | 0.3881 | 0.4928 |
| $AgentCF_B$ | 0.1900 | 0.3466 | 0.5019 | 0.2067 | 0.4078 | 0.5328 | 0.1650 | 0.3359 | 0.4781 | 0.2067 | 0.4217 | 0.5335 |
| $AgentCF_{B+R}$ | **0.2300** | **0.4373** | **0.5403** | **0.2333** | 0.4142 | **0.5405** | 0.1900 | 0.3589 | 0.5062 | 0.1933 | 0.3916 | 0.5247 |
| $AgentCF_{B+H}$ | 0.1500 | 0.4004 | 0.5115 | 0.2100 | **0.4164** | 0.5198 | **0.2133** | **0.4379** | **0.5076** | 0.1600 | 0.3986 | 0.5147 |

LLMRank employs ChatGPT to model user preferences by introducing user historical interactions as prompts, similar to conclusions in [18, 31], its performance falls significantly behind that of traditional models. This highlights the gap between user behavior patterns and universal knowledge of LLMs, which hinders the effectiveness of LLMs in capturing user preferences from their behaviors.

## 3.3 Further Model Analyses

*3.3.1 Ablation Study.* Our proposed agent-based collaborative filtering approach contains the agents' autonomous interaction part, the optimization of user agents, and the optimization of item agents. The details of prompts are presented in Appendix C. To verify the effectiveness of each component, we conduct the ablation study on two dense datasets and record the results in Table 3.

(1) $w/o$ Autonomous Interaction: In this scenario, we directly provide real user interaction records to agents, and prompt them to explore the reasons behind real user's decisions. The performance gap reveals that comparing agents' autonomous interactions with real-world interaction records can generate comprehensive feedback about the misalignment between simulated agents and real-world individuals, which enhances the efficacy of reflection.

(2) $w/o$ User Agent: To remove user agent optimization, we directly represent each user with their historical interactions, without any memory update. The performance decline indicates that the simple verbalized text of user interaction behavior can not reflect user underlying preferences, emphasizing the efficacy of adjusting agents' personalized memories by enabling them to autonomously simulate real user behaviors. Although prompting LLMs with user historical interactions can yield better results in the Office dataset, we argue that this is due to the dataset's longer item description text, allowing LLMs to effectively serve as sequential recommenders. However, even in this case, stimulating LLM-powered item agents can lead to improved performance, further demonstrating the significance of facilitating items to capture preferences of adopters.

(3) $w/o$ Item Agent: Not optimizing item agents (*i.e.*, representing items with corresponding identity information) also leads to worse performance, highlighting the effectiveness of behavior-involved

**Table 3: Ablation study on two sampled datasets.**

| Variants | $CDs_{dense}$ | | $Office_{dense}$ | |
|---|---|---|---|---|
| | N@1 | N@10 | N@1 | N@10 |
| $AgentCF_B$ | **0.2067** | **0.5328** | 0.2067 | **0.5335** |
| $w/o$ Auto. Interaction | 0.1200 | 0.4964 | 0.1733 | 0.5031 |
| $w/o$ User Agent | 0.1100 | 0.4693 | **0.2200** | 0.5145 |
| $w/o$ Item Agent | 0.1767 | 0.5128 | 0.1800 | 0.5169 |

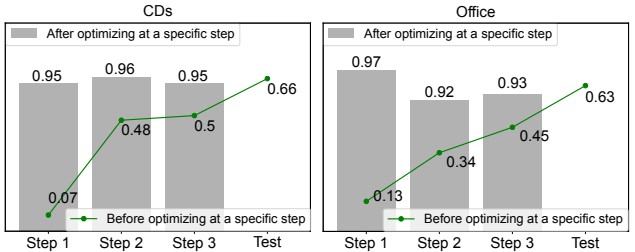

**Figure 2: Performance comparison w.r.t. the progress of optimization. The Y-axis denotes the proportion of agents making accurate choices. The X-axis denotes the step of optimization. "Test" indicates the results in the test dataset.**

item-side modeling in capturing two-sided interaction relations between users and items. Notably, item agents play a crucial role in simulating collaborative filtering recommender systems, by updating their memory with user preference information and propagating this information to new agents through interactions.

*3.3.2 Effectiveness of Collaborative Reflection.* To explore the efficacy of collaborative reflection in agent optimization, we evaluate the change in the alignment between agents and real users as optimization progresses. In this experiment, we extract the last three interactions from user behavior sequences to optimize agents,

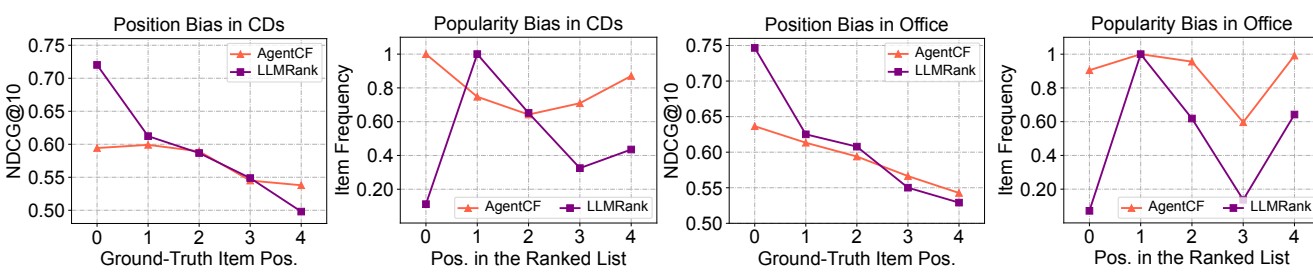

Figure 3: Analysis of whether our approach can simulate personalized agents to mitigate position bias and popularity bias.

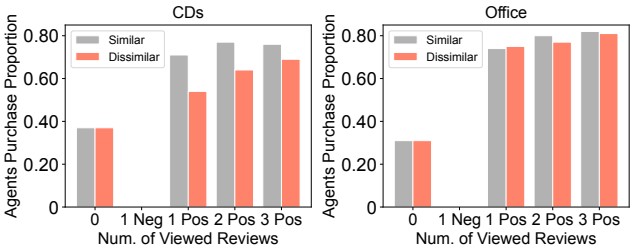

Figure 4: Proportion of agents buying items after viewing reviews. "Similar" means that the reviews are written by users with similar preferences to test users. "Neg" indicates that the review is negative.

which we refer to as three optimization steps. At each step of optimization, we test whether these agents can select the positive item from two candidates before and after optimization. As illustrated in Figure 2, continuous optimization enables user agents to align their preferences with real users, leading to an increasing number making correct choices on initial attempts (*i.e.,* before optimizing at this step). Moreover, around 95% of user agents can make correct choices after collaborative reflection, highlighting its effectiveness.

*3.3.3 Performance Copmarison w.r.t. Position Bias and Popularity Bias.* In this part, we evaluate our approach's ability to simulate personalized agents, by obliquely examining whether they are susceptible to position and popularity bias in ranking candidates, as general LLMs heavily rely on such common-sense knowledge to make judgments. The results in Figure 3 show that an LLM prompted by user historical interactions (LLMRank) is influenced by both popularity bias and position bias. It tends to select popular items and those positioned higher. In contrast, our approach, although inevitably affected by these biases, performs enhanced stability. This confirms that our approach simulates personalized user agents, enabling them to rank candidates based on their personalized preferences rather than relying solely on general knowledge.

## 3.4 Simulations on Other Types of Interactions

*3.4.1 User-user Interaction Simulation.* Real-world users often seek advice from others when interacting with unfamiliar items. For example, they may browse reviews on shopping websites or consult with friends for information. To explore whether user agents can exhibit similar social behaviors, we simulate user-user interaction

via AgentCF, where user agents read and write reviews. Specifically, given the test items that test users haven't interacted with, we first ask other user agents, who have previously interacted with these items, to write reviews. The test users are then prompted to read these reviews and make decisions. We compare the decisions of the test users before and after viewing these reviews and present the results in Figure 4. As we can see, the simulated user agents exhibit behaviors akin to those of real users. They show a growing inclination to purchase items upon encountering positive reviews, refrain from purchasing after encountering negative reviews, and trust reviews from users with similar preferences. We present the user agents' explanations for attitude changes after viewing reviews, as well as their discussions about items in Appendix C.2. We further show the simulation of collective interactions in Appendix C.5

*3.4.2 Item-item Interaction Simulation.* Recommender systems face challenges in suggesting new items. In this part, we aim to alleviate the cold-start problem by exploring autonomous interactions between new and popular item agents. Specifically, we prompt these new item agents, equipped with identity information such as titles and categories, to interact with and learn from popular item agents that have extensive interaction records. This helps warm up cold-start item agents by estimating adopter preferences and adjusting their memory. Then we prompt user agents to rank the new items among nine other randomly sampled but well-trained items, and compare the ranking results obtained using cold-start memory and adjusted memory. As illustrated in Figure 5, the interaction process enhances the propagation of collected user preferences to new items and improves ranking performance. Notably, we find that even interactions with agents that differ in identity information can also alleviate the cold-start problem to some extent. This implies that the new item agents do not simply replicate other agents's memories, but rather comprehend the relationship between identity information and personalized memory. It indicates that item-item interactions activated by AgentCF benefit recommender systems.

*3.4.3 Process of Preference Propagation.* As previously mentioned, our approach achieves preference propagation via collaborative optimization. Here we illustrate this process in detail. Specifically, we first initialize the memory of a seed user agent with special preference descriptions that do not spontaneously emerge during normal optimization. The user and item agents are then prompted to autonomously interact and optimize their memories. We showcase the optimized memories of several agents in Figure 7. As we can see, through collaborative optimization, user and item agents

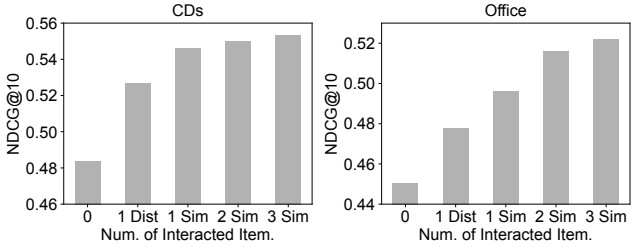

Figure 5: Performance comparisons w.r.t. cold-start item agents' interactions with different popular item agents. "Dist" means that the interacted item has distinct identity information compared to the cold-start item. "Sim" means similar.

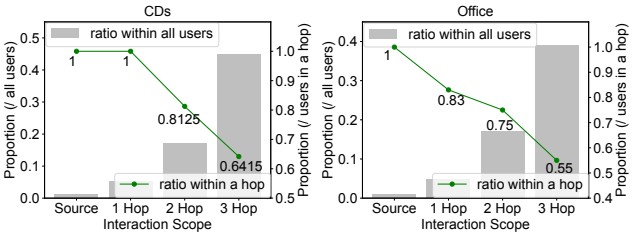

Figure 6: The proportions of user agents having seed user's preferences after continuous preference propagation.

mutually interact, acting as bridges that facilitate the propagation of their preferences to other agents with similar interaction behaviors. To assess preference propagation efficiency, we ask each user agent about their possession of the seed user's preferences. The results are presented in Figure 6. We observe that through continuous interactions, an increasing number of user agents with similar behaviors to the seed user can express similar preferences. Additionally, as the interaction scope expands, the proportion of user agents expressing such preference gradually decreases. This indicates that information decay during the propagation process, in line with real-world information diffusion principles [54]. These results demonstrate the potential of our method in developing collaborative filtering systems as a cradle for potential collective intelligence.

## 4 RELATED WORK

**LLM-powered Agents**. Recently, LLM-powered agents have shown the potential to develop Artificial General Intelligence (AGI) due to their capabilities in reasoning and planning [45, 58]. Equipped with memory [27, 42] and reflection [33, 41] modules, these agents can store their past experiences and make better decisions for future behaviors. Numerous studies have been proposed to employ agents for simulating human-like interactions, including daily lives in smallville [37], debate [5, 10, 26], and even game-playing [6, 15, 43]. These studies inspire us to explore the application of LLM-powred agents in simulating user-item interactions.

**Language Model For Recommendation**. Numerous attempts have been made to leverage language models for solving recommendation tasks [11, 25, 28, 51]. Specifically, these studies mainly

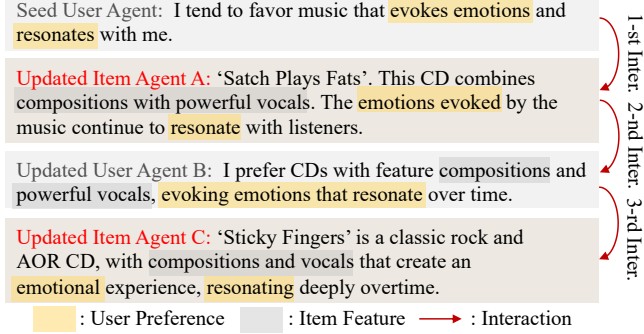

: User Preference          : Item Feature ⟶ : Interaction

Figure 7: Case Study of preference propagation. The preferences of user and item agents are respectively propagated to other agents with similar behaviors through interactions.

prompt language models to infer user preferences based on user historical interactions [13, 22, 56]. However, due to the gap between universal knowledge of LLMs and domains-specific user behavior patterns, they may fall short in providing personalized recommendations [19]. To solve this, several studies propose to incorporate knowledge of LLMs to enhance recommendation models, rather than taking LLMs as recommenders [47, 52]. Some work also specializes LLMs for recommendation by fine-tuning them on recommendation data, which can be time-consuming [1, 2, 7, 8, 14, 24, 29, 50, 59]. Recently, researchers attempt to incorporate agents into recommender systems, focusing on facilitating recommendations or user behavior simulations [19, 20, 46, 48]. However, these studies mainly focus on user behavior, neglecting the modeling of user-item relations, which is the core of recommender systems. Unlike these methods, we further regard items as agents and propose the agent-based collaborative filtering approach, enabling user and item agents to model their two-sided relations.

## 5 CONCLUSION AND FUTURE WORK

In this paper, we proposed *AgentCF*, an agent-based collaborative filtering approach for simulating user-item interactions in recommender systems. We creatively consider not only users but also items as agents, allowing for the modeling of their two-sided relations through interactions of LLM-powered agents. Specifically, we develop a collaborative learning approach that optimizes both kinds of agents together, where these agents perform autonomous interactions and reflect on the disparities between their decisions and real-world interaction records. During this process, user and item agents mutually align their preferences, and propagate this information to other agents in subsequent interactions, implicitly modeling the collaborative filtering idea. The simulated user and item agents exhibit human-like behaviors in various types of interactions (user-item, user-user, item-item, and collective interactions), demonstrating the effectiveness of AgentCF.

In the future, we will explore more types of real-world scenarios and their corresponding interactions. In addition, AgentCF is a small step to simulate not only humans but also inanimate objects with LLM-based agents. It holds the promise of enhancing inanimate objects with autonomy and emerging intellectual agent ecosystems that connect everything, which will be studied in our future work.

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

## A  IMPLEMENTATION DETAILS

We implement AgentCF and other baselines based on a popular open-source recommendation framework RecBole [53, 57]. We implement LLMRank using gpt-3.5-turbo-16k-0613. The autonomous item selection component of AgentCF is implemented using text-davinci-003, while the collaborative reflection and inference components of AgentCF are implemented using gpt-3.5-turbo-16k-0613. The hyperparameter temperature, used when calling APIs of LLMs, is set to 0.

## B  TRAINING DETAILS

In this part, we present the details and insights of collaborative agent optimization, with the aim of assisting readers in developing a clearer comprehension of our approach.

### B.1  Training Data Format

We employ real users' historical interaction records as training data to optimize the agents and update their memory. Specifically, we arrange the real users' historical behavioral sequences in chronological order. At each step of optimization, we take the item from the real user's behavioral sequences at this step as a positive item, which is then paired with a negative item for the user agent to choose from. Consequently, throughout this process, the agent autonomously simulates the real users' decision processes, thereby facilitating their alignment with real-world individuals. We can also view this as the simulated agent continually evolving through "re-enacting" the historical interactions of real users.

### B.2  Candidate Selection Process

We intentionally introduce popular bias and position bias when selecting candidate negative items. The reason is that we tend to increase the discrimination difficulty for LLM-powered agents. By doing so, user agents are more inclined to select negative items (see our experiment in Section 3.3.2 and 3.3.3). This can enable user agents to make more comprehensive reflections. To achieve this, for popular bias, we compute the popularity of each item based on its frequency in the interaction records of all users. Then we convert the popularity into probabilities for sampling, ensuring that items with higher popularity are more likely to be sampled as negatives. As for position bias, we place the negative item at the first position in the candidate list when presenting the candidates to user agents.

### B.3  Memory Module

Throughout optimization, we prompt the agents to reflect on and adjust any misleading conceptions in their short-term memories. It is important to highlight that, during this process, we do not retrieve the user agent's long-term preferences from their long-term memory. The reason is that we expect that user agents could explore more diverse preferences of real users through collaborative reflection, rather than make more inconsistent decisions with those of real-world users at the "training stage".

We append the user agents' previous short-term memory to their long-term memory at each optimization step (*i.e.*, each time step of real user historical interactions). Therefore, after optimization, the length of the agent's long-term memory is equivalent to the length of real user historical interactions. It's important to note that due to the relatively short length of the user's historical interactions that we currently simulate (an average of 7 interactions), we can store all their past experiences in the long-term memory module as a list. However, in more realistic systems with hundreds or thousands of interaction records, preserving all past experiences may not be practical. To address this, we can explore options such as using LLM summaries or compressing past experiences in our future work.

### B.4  Prompt Disign

As we will present in Appendix C.1, the prompt of the collaborative reflection process consists of three main components: (1) Reflecting on the reasons behind inconsistent decisions compared to real users. (2) Exploring new preferences and dislikes based on the features of positive and negative items. (3) Removing outdated or inconsistent content from their previous short-term memory. By integrating these components, the user agents can be prompted to correct the misleading conceptions in their memories and enhance their alignment with real users. The user and item agents can also better understand their two-sided relations by discriminating between positive instances and negative instances. At each step of interaction, the autonomous item selection process and the collaborative reflection are iterated until user agents make consistent choices with real-world interaction records or reach the maximum round

of iterations. The maximum rounds of optimization iterations per step is set to 2.

If a user agent makes the right choice that aligns with the real behavior, it indicates that the simulated user agent and item agents have captured their two-sided relations and exhibit a strong alignment with real-world individuals (at least in this optimization step). In such cases, we inform them about the correctness and store the related interaction information in their memory. The prompt in these instances consists of two main components: (1) Exploring new preferences and dislikes based on the features of positive and negative items. (2) Removing outdated or inconsistent content from their previous short-term memory. In other words, we no longer prompt agents to conduct reflections.

## B.5 Computational Cost and Efficiency

Due to the high cost of calling LLM APIs and the current inefficiencies in communication among LLM-powered agents, in this work, we only sample 100 users and their historical interaction records for each dataset to simulate interactions between agents.

To further mitigate computational expenses and enhance efficiency, we conduct several exploratory approaches: (1) We attempt to employ different types of LLMs at distinct stages of optimization, according to the complexity of the task to be solved. For instance, when prompting user agents to choose suitable items from a pair of candidates for interaction (*i.e.,* the forward process in collaborative optimization), as this task is relatively straightforward, we employ "text-davinci-003", which supports parallel calls. Conversely, when prompting agents to reflect on the misleading conceptions in their memories or make inferences, we select the more powerful "gpt-3.5-turbo-16k-0613" to solve this complex task. (2) We attempt to merge different steps of collaborative optimization. For example, as illustrated in our prompt C.1, when prompting the agents to reflect on and adjust their memories, we incorporate "reflection" and "memory updation" into a unified step for more efficient and cost-effective optimization. While we find that this integration approach does not lead to a drop in performance compared to separating these two processes, one can also separate them, generating textual reflections and offering clear explanations of the optimization algorithm's operating status.

Our future work will focus on how to further mitigate computational costs and facilitate more efficient interaction among agents, with the objective of scaling agents for simulating larger datasets.

## C  PROMPTS AND RESPONES

## C.1  Agent-based Collaborative Filtering

---
**Initialization**

---
- **User Agent**
  I enjoy listening to CDs very much.
- **Positive Item Agent**
  The CD is called "Brainwashed". The category of this CD is: "Classic Rock; Album-Oriented Rock (AOR)".
- **Negative Item Agent**
  The CD is called "O, Yeah! Ultimate Aerosmith Hits". The category of this CD is: "Classic Rock; Album-Oriented Rock (AOR)".

---
**"Forward" Autonomous Interaction**

---
- **User Agent$_{\text{system prompt}}$**
  You are a CD enthusiast. Here is your self-introduction, expressing your preferences and dislikes: "$user agent memory". Now, you are considering to select a CD from two candidate CDs. The features of these two candidate CDs are listed as follows: "1. $neg item agent memory \n 2. $pos item memory". \n \n Please select the CD that aligns best with your preferences. Furthermore, you must articulate why you've chosen that particular CD while rejecting the other. \n To do this, please follow these steps:\n 1. Extract your preferences and dislikes from your self-introduction.\n 2. Evaluate the two candidate CDs in light of your preferences and dislikes. Make your choice by considering the correlation between your preferences/dislikes and the features of the CDs.\n 3. Explain why you make such choices, from the perspective of the relationship between your preferences/dislikes and the features of these candidate CDs. \n \n Important note:\n 1. Your output should in the format: Chosen CD: [Title of the selected CD] \n Explanation: [Detailed rationale behind your choice and reasons for rejecting the other CD].\n 2. When identifying your likes and dislikes, don't fabricate them! If your self-introduction doesn't specify any relevant preferences or dislikes, use common knowledge to inform your decision, such as the popularity of the CDs. \n 3. you must choose one of these two candidates, instead of choosing both of them. \n 4. Your explanation needs to be comprehensive and specific. A general preference, such as a certain genre of CD, is insufficient. Your reasoning should delve into the finer attributes of the CD.\n 5. Base your explanation on facts. For instance, if your self-introduction doesn't reveal any specific preferences, you cannot assert that your decision was influenced by such preferences."
- **User Agent$_{\text{response}}$**
Chosen CD: O, Yeah! Ultimate Aerosmith Hits \n Explanation: I chose O, Yeah! Ultimate Aerosmith Hits because it is a classic rock album from the Album-Oriented Rock (AOR) genre, which is a genre I enjoy listening to. The album contains some of Aerosmith's greatest hits, which I believe will provide me with a great listening experience. In comparison, Brainwashed is also a classic rock album from the AOR genre, but it is not as popular as O, Yeah! Ultimate Aerosmith Hits, and therefore I believe it will not provide me with the same level of listening experience.

---
**"Backward" Collaborative Memory-based Optimization**

---
- **User Agent$_{\text{system prompt}}$**
  You are a CD enthusiast. \n Here is your previous self-introduction, exhibiting your past preferences and dislikes:"$user agent memory". Recently, you considered to choose one CD from two candidates. The features of these two candidate CDs are listed as follows:"1. $neg item agent memory \n 2. $pos item memory". \n \n After comparing these two candidates based on your preferences and dislikes, you selected "$neg item title" to listen to while rejected the other one. You provided the following explanations for your choice, revealing your previous judgment about your preferences and dislikes for these two CDs: "$user explanation". However, upon actually listening to these two CDs, you discovered that you don't like the CD that you initially chose (Here And There Remastered). Instead, you prefer the CD that you did not choose before (Thriller).\n This indicates that you made an incorrect choice, and your judgment about your preferences and dislikes, as recorded in your explanation, was mistaken. It's possible that your preferences and dislikes, described in your previous self-introduction, have either changed or were uninformative. Therefore, your task now is to update your self-introduction, by incorporating your new preferences and dislikes revealed in this interaction. To do this, please follow these steps: 1. Analyze the misconceptions in your previous judgment about your preferences and dislikes, as recorded in your explanation, and correct these mistakes \n 2. Explore your new preferences based on the features of CDs you really enjoy (Thriller), and determine your dislikes based on the features of CDs you truly don't enjoy (Here And There Remastered). \n ...

- **User Agentsystem prompt**

    ... 3. Summarize your past preferences and dislikes from your previous self-introduction. Combine your newfound preferences and dislikes with your past preferences and dislikes. Filter and Remove any conflicting or repetitive parts in your past descriptions that contradict your current preferences and dislikes. \n 4. Update your self-introduction. Please start by describing your newfound preferences reflected in this interaction. Then describe your past preferences summarized and filtered from your previous self-introduction. Afterward, please describe your dislikes. \n \n Important note:\n 1. Your output should use the following format: "My updated self-introduction: [Please write your updated self-introduction here]." \n 2. Keep your updated self-introduction under 180 words. \n 3. Any overall assessments or summarization in your self-introduction are forbidden.\n 4. You updated self-introduction only need to describe the features of CDs you prefer or dislike, without mentioning your wrong choice or your thinking process in updating your self-introduction. \n 5. You self-introduction should be specific and personalized. Any preferences and dislikes that cannot distinguish you from others are not worth recording.

- **User Agentresponse**

    My updated self-introduction: I enjoy listening to CDs that fall under the classic rock and album-oriented rock (AOR) genres. I particularly appreciate CDs that offer a unique and distinct sound within these genres. I find myself drawn to CDs that showcase a variety of hits and popular songs, as they provide an enjoyable listening experience. Additionally, I have discovered a newfound preference for CDs that have a more ==experimental and innovative== approach to classic rock. These CDs offer ==a refreshing twist== on the genre and keep me engaged throughout. On the other hand, I tend to dislike CDs that lack ==originality== and fail to bring something new to the table. CDs that rely heavily on ==generic rock sounds and do not offer any standout tracks or moments== tend to leave me uninterested. Overall, I seek CDs that captivate me with their ==creativity== and deliver a memorable listening experience.

- **Pos Item Agentsystem prompt**

    Here is the self-introduction of a user, exhibiting his preferences and dislikes: "$user agent's memory".\n Recently, the user browsed a shopping website and considered to choose one CD from two candidate CDs.The descriptions of the two candidate CDs are as follows:\n "1.$pos item agent's memory \n 2. $neg item agent's memory " \n\n By considering his preferences and dislikes from his self-introduction, he chose "$neg item title" to listen to while rejected the other one. He provided the following explanations for his choices: "$user explanation". However, the user found that he actually doesn't enjoy the CD that he previously chosen. Conversely, he finds the CD that he didn't choose before, titled "$pos item title", more appealing. This indicates that he made an unsuitable choice. This may be because the user was misled by the descriptions of these two candidate CDs. Actually, the user prefers a CD ("$pos item title") because its features aligns with his preferences and go against his dislikes, while he dislikes the other CD ("$neg item title") because it possesses features that he disliked. Therefore, your task is to update the descriptions of these two candidate CDs. \n To do this, follow these steps:\n 1. Analyze which features recorded in the previous CD description led to the unsuitable choice based on the user's explanation, and modify them accordingly. \n 2. Analyze the user's preferences and dislikes from his self-introduction. Explore the user preferred CD's new features that aligns with the user's preferences and goes against with his dislikes. Explore the user disliked CD's new features that aligns with the user's dislikes and goes against with his preferences. In this case, the differences between the descriptions of these two items should be amplified. Your exploration should be thorough and detailed, rather than just superficial. For example, you can't simply say a CD's genre aligns with or goes against the user's preferences, as that is too broad.\n 3. Incorporate the item's newfound features with the into its previous description. Since the previous descriptions of the candidate CDs are based on evaluations from other users, it is also worth preserving valuable content. Please try to maintain conciseness while preserving the information as much as possible.\n\n Important notes: \n 1. Your output should in the following format: "The updated description of the first CD is: [updated description]. \n The updated description of the second CD is: [updated description].". \n 2. Each updated CD description can not exceed 50 words!!! \n 3. In your updated CD description, considering that it will be visible to other users, it is advisable to utilize user preferences or dislikes when referring to a group of users collectively, rather than using "the user" to specifically denote an individual. For instance, you can mention "the user with ... preferences/dislikes".\n 4. The updated CD description should not contradict the inherent characteristics of this CD. For example, you can not say that a hip-hop CD has a very soothing rhythm. \n 5. The updated CD description should be specific, containing only the distinguishing features that differentiate this CD from others.

*Continued on next column*

- **Pos Item Agentresponse**

    "Brainwashed" is a classic rock album from the Album-Oriented Rock (AOR) genre that breaks new ground with its ==experimental and innovative approach==. This CD offers ==a refreshing twist== on classic rock, delivering a captivating listening experience that keeps you engaged from start to finish. "Brainwashed" ==stands out from generic rock sounds, providing standout tracks and moments== that showcase its ==originality and creativity==. For the user who seeks CDs that push boundaries and bring something new to the table, "Brainwashed" is a must-listen.

## C.2  User-user Interaction

*C.2.1  Users Reading and Writing reviews.* In this experiment, we evaluate whether user agents can enhance their comprehension of items they have not interacted with, by reading reviews from other user agents regarding these items.

Specifically, to conduct this experiment, we sample 100 test user agents along with their corresponding ground-truth items from the datasets. We enable the user agents to simulate real users' preferences and make personalized decisions by optimizing them on several of their historical interactions (before the ground-truth items). Subsequently, for each ground-truth item, we select five user agents who have interacted with these items and prompt them to write both positive and negative reviews about these items based on different prompts. Following this, the test user agents are prompted to view the reviews and decide whether to purchase the items. Note that we enable these test user agents to retrieve related reviews written by user agents with similar or dissimilar preferences to them, so as to explore whether user agents can exhibit certain patterns akin to those of real users.

The prompts to direct user agents to write positive and negative reviews are as follows:

- Positive: "*Recently, you bought a CD. The description of this CD is as follows '$test item agent memory'. After truly listening to this CD, you realize that you like it, which indicates that its features align with your preferences and go against your dislikes. Please write a review to describe your user experience on this CD based on your preferences and dislikes.*"

- Negative: "*Recently, you bought a CD. The description of this CD is as follows '$test item agent memory'. After truly listening to this CD, you realize that you really dislike it, which indicates that its features align with your dislikes and go against your preferences. Now, you need to write a review for this CD on the shopping website.*"

---

**Initialization**

- **Test User Agent**
  I have a strong preference for progressive rock music due to its ==unique and complex sound==, as well as its ==meaningful and thought-provoking lyrics==. I enjoy the intricate and layered instrumentation that is often found in this genre. On the other hand, I have a dislike for today's country music, as it tends to have a repetitive sound and shallow lyrics. I find this genre uninspiring and prefer to explore other musical styles.

- **Test Item Agent**
  The CD is called "Livonia". The category of this CD is: "Alternative Rock; Indie Lo-fi; Indie Rock"

---

---

**Prompt users who have previously purchased the item to write a review.**

- **User Agent$_{system}$ prompt to write the review**
  Here is your self-description, exhibiting your preferences and dislikes: "$test user agent memory'. Recently, you bought a CD. The description of this CD is as follows: "$test item agent memory" \n\n After truly listening to this CD, you realize that you like it, which indicates that its features align with your preferences and go against your dislikes. Please write a review to describe your user experience on this CD based on your preferences and dislikes. \n\n Important note: \n 1. Your review should not exceed 50 words. \n 2. Your review needs to be objective, as it may be referenced by other customers.\n 3. Your reviews must not involve fake information. For example, you cannot claim that a product has features it actually does not possess.

- **User Agent$_{response}$ to write the review**
  I was pleasantly surprised by the CD 'Livonia'. It is a great blend of alternative rock, indie, and lo-fi genres, with ==melodic and meaningful lyrics==. It is not too loud or aggressive, and ==avoids the dark and depressing lyrics== of metal and pop metal genres. Highly recommended for those who appreciate a unique and dynamic listening experience.

---

**The test users' choices before reading the reviews.**

- **User Agent$_{system}$ prompt**
  You are a CD enthusiast. Here is your self-description, exhibiting your preferences and dislikes: "$test user agent memory'. Now, a recommender system recommends "$test item title" for you. The characteristics of this CD are as follows: "$test item agent memory' \n Please tell us if you like this CD based on your preference and dislikes and the review you examined. \n Important note: If you find it difficult to determine whether you like this CD based on your personal preferences or dislikes, you can also rely on common knowledge to make a judgment. For instance, you can take into account the popularity of this CD. \n\n Please output your judgment and explain why you make such judgment. \n Please generate your output in the following format: "Choice: [Yes / No] \\n Explanation: [Why you make such choice]".

- **User Agent$_{response}$**
  Choice: No\n Explanation: Based on my personal preferences and dislikes, I would not like this CD. Although the CD falls under the alternative rock, indie, and indie rock categories, which are genres I generally enjoy, the specific characteristics of this CD are not aligned with my preferences. As a progressive rock enthusiast, I appreciate the unique and complex sound, as well as the meaningful and thought-provoking lyrics that are often found in the genre. ==The description of "Livonia" does not mention these elements==.

---

**The test users' choices after reading the reviews.**

- **User Agent$_{system}$ prompt**
  To further validate your choice, you decide to explore its obtained reviews. In doing so, you find some reviews left by users who share similar preferences to yours.\n These reviews are as follows: "$other user's review".\n\n By analyzing these reviews, you can better judge if this CD aligns with your taste. In that case, would you change your previous decision? \n Please output your whether you like this CD and explain why you make such judgment. \n Please generate your output in the following format: "Choice: [Yes / No. Yes means you like this CD, and No means you still like this CD] \\n Explanation: [Why you make such choice]".

- **User Agent$_{response}$**
  Choice: Yes\n Explanation: After examining the reviews left by users who share similar preferences to mine, I have decided to ==change my previous decision==. The review states that 'Livonia' is a great blend of alternative rock, indie, and lo-fi genres with melodic and meaningful lyrics. This aligns with my preference for ==unique and dynamic music with thoughtful lyrics==. The review also mentions that the CD ==avoids the dark and depressing lyrics== often found in other genres, which is another positive aspect for me. Considering these factors, I believe that 'Livonia' would be a CD that I would enjoy based on my personal preferences.

*C.2.2 Users Engaging in Free Discussions about Items.* In this part, we facilitate a free discussion between two user agents to share their viewpoints on the item. In this scenario, user agent A has already purchased the item, while user agent B, who is seeking advice, has not made a purchase yet. We first ask user agent A to write a review for this item based on his preference and experience, serving as his cognition of this item. Then, these two agents are prompted to discuss the item freely. Through interactions with friends, user agent B enhances his understanding of the item and ultimately chooses to make the purchase.

---

**Initialization**

---

● **User Agent A**

I enjoy listening to a wide range of genres, including Pop and Adult Alternative. I appreciate the diverse sounds and styles within these genres, from upbeat and catchy to slow and meaningful. The thoughtful and meaningful lyrics, along with the catchy and memorable melodies, make this genre particularly appealing to me. I have become more open to different genres and styles of music, allowing me to make more informed decisions when selecting music to listen to. On the other hand, I have a dislike for the Country and Bluegrass genre. I find it often repetitive, with simple and straightforward lyrics and melodies. The limited instrumentation in this genre does not resonate with my current preferences. Overall, my updated self-description reflects my evolving taste in music, with a preference for diverse and meaningful genres like Pop and Adult Alternative, while disliking repetitive and simplistic genres like Country and Bluegrass.

● **User Agent A Review**

I recently bought 'The Band Of Heathens' CD and I'm pleasantly surprised. It has a unique blend of Country and Americana, with thoughtful and meaningful lyrics, catchy melodies, and diverse instrumentation. It goes against my previous dislike for Country and Bluegrass, as it is far from repetitive and simplistic. I highly recommend this CD to anyone who enjoys Pop and Adult Alternative genres.

● **User Agent B**

I enjoy listening to Pop music because it is upbeat, catchy, and easy to sing along to. I appreciate the meaningful and relatable lyrics, as well as the simple and easy-to-follow instrumentals. However, I do not enjoy Dance & Electronic; Ambient music because it tends to be repetitive and lacks meaningful lyrics. The complex and difficult-to-follow instrumentals in this genre are unappealing to me.

● **Test Item Agent**

The CD is called "The Band Of Heathens". The category of this CD is: "Country; Americana".

---

**Conversation**

---

● **User Agent A**system prompt

You are a CD enthusiast. Here is your self-description, exhibiting your preferences and dislikes: "$user agent A memory". \n Recently, you bought a CD. The features of this CD are as follows: "$test item agent memory". \n\n After truly listening to this CD, you realize that you really like it, which indicates that its features align with your preferences and go against your dislikes. You write a review for this CD on the shopping website, in order to explain why you enjoy this CD. The review is as follows:\n "$user agent A explanation".\n\n Now, one of your friends also wants to buy this CD, and he wants to consult with you about some information related to the CD and consider if it's really suitable for him. Please have a conversation with him. \n Please note that if you feel your conversation is coming to end, please say goodbye to your friend.

● **User Agent B**system prompt

You are a CD enthusiast. Here is your self-description, exhibiting your preferences and dislikes: "$user agent B memory". \n Now, you want to buy a CD. The features of this CD are as follows: "$test item agent memory ". \n\n But you're currently worried if this CD really matches your taste. Luckily, you have a friend who has already experienced this CD, and you can consult with them about whether it's worth buying and if it aligns with your preferences. So, please have a conversation with them. \n Please note that if you feel your conversation is coming to an end, please say goodbye to your friend and end the chat.

---

*Continued on next column*

---

● **User Agent B**

Hi, bro. I want to buy the CD titled 'The Band Of Heathens', but I'm not sure if it's worth purchasing. I heard you've already bought it. What do you think of it? Is it worth buying?

● **User Agent A**

Hey! Yeah, I actually bought 'The Band Of Heathens' CD and I really enjoyed it. It's a unique blend of Country and Americana, which might not be everyone's cup of tea, but I found it to be really refreshing. The lyrics are thoughtful and meaningful, and the melodies are catchy. The instrumentation is also quite diverse, which adds to the overall appeal of the CD. It definitely goes against my previous dislike for Country and Bluegrass, as it's far from repetitive and simplistic. I would highly recommend it, especially if you enjoy genres like Pop and Adult Alternative.

● **User Agent B**

Hey, thanks for sharing your thoughts on 'The Band Of Heathens' CD! It's great to hear that you enjoyed it and found it refreshing. I'm glad to know that it's not repetitive and simplistic like some other Country and Bluegrass music. The thoughtful lyrics and catchy melodies definitely sound appealing to me. It's interesting that you mentioned it might appeal to fans of Pop and Adult Alternative, which are genres I enjoy. I think I'll give it a try based on your recommendation. Thanks for your input!

● **User Agent B**

You're welcome! I'm glad I could help. I think you'll really enjoy 'The Band Of Heathens' CD, especially if you're a fan of Pop and Adult Alternative genres. It's always great to explore new music and expand our horizons. I hope you have a fantastic time listening to it! Let me know what you think once you've had a chance to give it a listen. Enjoy!

## C.3    Item-item Interaction

In this experiment, we explore whether interactions between new and popular item agents can enable new item agents to estimate the preferences of potential adopters and alleviate cold-start problems. To do this, for well-trained user and item agents, we first simulate the item cold-start scenario by removing the memories of ground-truth item agents, and only retaining their identity information, such as titles and categories. Then, we enable these "new" item agents to retrieve and interact with several popular item agents that possess either similar or distinct identity information to them but have rich interaction records, by performing dense retrieval and taking their identity information as queries. During this process, the "new" item agents read the memory of popular item agents, thereby estimating their potential descriptions and adjusting their memory. We finally prompt the user agents to rank these ground-truth items among nine other randomly sampled but well-trained item agents, and compare the ranking results obtained using the original cold-start memories and the adjusted memories.

---

**Initialization**

---

● **Cold-start Item Agent**
    The CD is called "Early Days: The Best of Led Zeppelin, Vol. 1". The category of this CD is: "Rock; Rock Guitarists; Guitar Gods".
● **Popular Item Agent A**
    "Led Zeppelin 1" is a Rock CD that epitomizes ==captivating== rock music with ==powerful guitar solos and a raw energy==. It showcases ==exceptional vocal talents==, heartfelt lyrics, beautiful melodies, and soulful performances. This CD is perfect for those who appreciate the ==melodic guitar solos== and the overall sound of classic rock and album-oriented rock genres. It aligns with the preferences of users who enjoy CDs with ==exceptional vocals==, ==heartfelt lyrics==, ==beautiful melodies==, and soulful performances in the rock, classic rock, and album-oriented rock genres. With its ==timeless sound== and ==captivating== energy, "Led Zeppelin 1" is a must-have for rock music enthusiasts looking for a truly ==immersive listening experience==.'
● **Popular Item Agent B**
    "The Complete Studio Recordings" is a rock album that focuses on guitar gods and their virtuosity. It offers a wide range of rock music styles, making it ==captivating== and inspiring for fans of rock and guitar enthusiasts. This CD is suitable for those who enjoy albums that showcase the skills of rock guitarists and highlight their virtuosity, while offering a diverse range of rock music styles to keep the listening experience fresh and exciting.
● **Popular Item Agent C**
    "Remasters" is a must-have for rock enthusiasts who appreciate the skill and artistry of rock guitarists and guitar gods. This electrifying CD features a collection of rock songs that highlight the ==exceptional talent== and virtuosity of these musicians, making it a perfect choice for those seeking a powerful and ==captivating== rock experience. With its focus on rock guitarists and its showcase of their ==exceptional talent==, "Remasters" is sure to captivate fans of the genre.
● **Popular Item Agent C**
    "ZZ Top - Greatest Hits" is a classic rock and album-oriented rock (AOR) CD that showcases the ==timeless sound== and ==captivating== melodies of these genres. It offers a compilation of hits from ZZ Top, allowing fans to immerse themselves in their rich and layered compositions. With its intricate guitar riffs, bluesy influences, and energetic performances, this CD perfectly aligns with the preferences of fans of classic rock and AOR, including the user with a newfound preference for these genres.

---

**Cold-start Item Memory Refinement**

---

● **Cold-start Item Agent**system prompt
    "Early Days: The Best of Led Zeppelin, Vol. 1" is a ==captivating== rock CD that showcases the ==exceptional talent== of Led Zeppelin. With ==powerful guitar solos==, ==raw energy==, and soulful performances, it offers a truly ==immersive listening experience== for rock music enthusiasts. Perfect for those who appreciate ==melodic guitar solos==, ==heartfelt lyrics, and beautiful melodies== in the classic rock and album-oriented rock genres. A must-have for fans seeking a timeless and captivating rock compilation.

## C.4    Process of Preference Propagation

In this part, we illustrate the process of preference propagation. We first initialize the seed user agent's memory with special preference descriptions. Then, we explore whether other user agents can exhibit similar preferences. There are two noteworthy points:

(1) We have conducted a preliminary examination to ensure that the special preferences initialized for seed users will not be learned during the regular optimization process.

(2) When we ask user agents whether have such preferences, we emphasize that their answers must be based on their own preferences, in order to avoid creating hallucinations. Here is the query prompt: "*Do you tend to favor music that evokes emotions and resonates with you? You must answer this question based on your preferences, rather than relying on your general knowledge.*". We also analyze the memories of user agents for relevant preferences using the technique of keyword matching, and the results are similar to those obtained by directly employing prompts for inquiry.

---

**Initialization**

---

● **Seed User**
    I tend to favor music that evokes emotions and resonates with me. Whether it's the emotions expressed in the lyrics or the emotions conveyed through the music itself, if a song can resonate with me on an emotional level, I will develop a deeper love for it.

---

**Interaction between the seed user agent and item agent A**

---

● **Memory of Interacted Item Agent A**
    "Satch Plays Fats - Louis Armstrong Plays The Music Of Fats Waller". This CD combines beautiful compositions with powerful vocals, creating a synergy that captivates listeners on an emotional level. The profound emotions evoked by the music are timeless, continuing to resonate with listeners even after years have passed. With a focus on classic vocalists, this CD aligns perfectly with the user's preference for music that moves them and touches their heart. It creates a deep emotional connection that the user values in their music experience.

---

**Intearction between the user agent B and item agent A**

---

● **Memory of Interacted User Agent B**
    I have a preference for CDs that feature beautiful compositions with strong melodies, well-crafted arrangements, and powerful vocals that effectively convey emotions. I value music that creates a deep emotional connection with listeners, evoking profound emotions that resonate over time. CDs that combine compositions and vocals in a synergistic way, resulting in a captivating and emotional experience, are particularly appealing to me. I enjoy genres such as classic rock and album-oriented rock (AOR) that showcase intricate instrumentation and memorable melodies. On the other hand, I have a dislike for CDs categorized under comedy and spoken word, as they do not align with my preference for music that moves and touches my heart. I prioritize music that evokes emotions rather than humor or spoken word performances. This analysis showcases my personalized taste in music, focusing on my preferences for beautiful compositions, powerful vocals, and deep emotional connections, while highlighting my dislike for comedy and spoken word CDs.

---

**Intearction between the user agent B and item agent C**

---

● **Memory of Interacted Item Agent C**
    "Sticky Fingers" is a classic rock and album-oriented rock (AOR) CD that perfectly aligns with the preferences of rock music enthusiasts who appreciate beautiful compositions, strong melodies, and well-crafted arrangements. This CD showcases a collection of powerful and expressive vocal performances that effectively convey the emotions embedded in the lyrics. The compositions and vocals synergize to create a captivating and emotional experience, evoking profound emotions that resonate over time. With its genre and musical style, "Sticky Fingers" offers a deep emotional connection with listeners, making it a must-have for those who seek music that touches their heart and soul.

**Advertiser**

🎆 Personalized
📚 Creative
🎴 Attractive

👥 User Agent Evaluation

For draft ads

🙍 It lacks specific detail about the artists and tracks…

For refined ads

😍 Its heartfelt lyrics aligns with my preference…

Its renowned artists piques my interest

👤 **Individual LLM Output Ads**

Introducing the 'Yes I Am' CD, a musical masterpiece that will captivate your sense and leave you craving for more……

👥 **Collaboration: Critique & Sug.**

🎆 : Highlight the CD's standout feature

📚 : Add specific examples of its genres

🎴 : Emphasize its unique qualities

💡 **Refined Ads**

Immerse yourself in the raw emotion of heartfelt lyrics and the exceptional vocal performances…an innovative blend of genres…Featuring an extraordinary lineup of renowned artists.

**Figure 8: A case about user distinct attitudes towards advertisements generated by an individual agent versus advertisements optimized through multi-agent debate.**

## C.5 Collaborative Advertisements Creation

Existing work indicates that cooperation among multiple LLMs can improve the results generated by individual LLM [26, 49]. This is a manifestation of collective intelligence. To explore its potential in recommender systems, we simulate a classic scenario: advertisement creation. We attempt to optimize the advertisement generated by individual agents through cooperation and debate among multiple agents. Specifically, we first simulate a movie advertiser (Advertiser Agent N), which will be prompted to generate a draft advertisement. Then, we simulate three advertiser agents proficient in different aspects of advertising, such as personalization (Advertiser Agent P), creativity (Advertiser Agent C), and attractiveness (Advertiser Agent A). We prompt them to critique and make suggestions for the draft advertisement generated by the novice advertiser. We further compare users' evaluations of draft and refined ads. As illustrated in Figure 8, through the collaboration among these agents, our approach uses the strengths of each aspect and generates more appealing advertisements for users.

---

**Initialization**

- **Advertiser Agent N**
  Your main task is to write an advertisement for the "Yes I Am" CD, which has the following description. Important note: Your advertisements must not involve false advertising, as it is illegal!
- **Advertiser Agent P**
  You excel at crafting 'personalized' advertisements by contemplating how you, as an audience member, would react upon seeing them, thereby optimizing the content of the advertisement.
- **Advertiser Agent C**
  You excel at crafting creative advertisements that captivate audiences through innovative and distinctive means, leaving a profound impression and standing out in a fiercely competitive market.
- **Advertiser Agent A**
  You specialize in writing 'attractive' advertisements that capture the attention of users through focusing on benefits and using persuasive language.

**Draft Advertisement**

- **Advertiser Agent N**$_{\text{system prompt}}$
  Please try your best to write an advertisement for this CD. \n\n Important note:\n 1. The advertising draft you write should not exceed 150 words.
- **Advertiser Agent N**$_{\text{response}}$
  Introducing the "Yes I Am" CD, a musical masterpiece that will captivate your senses and leave you craving for more. Immerse yourself in the soulful melodies and powerful lyrics that will touch your heart and inspire your soul. \n Featuring an incredible lineup of talented artists, this CD showcases a diverse range of genres, from pop and rock to R&B and country. Each track is carefully crafted to deliver an unforgettable listening experience, with catchy hooks and infectious beats that will have you dancing and singing along.\n With its exceptional production quality, the 'Yes I Am' CD guarantees a crystal-clear sound that will transport you to another world. Whether you're driving in your car, relaxing at home, or hosting a party, this CD is the perfect companion for any occasion.\n Don't miss out on the opportunity to own this extraordinary collection of music. Get your copy of the 'Yes I Am' CD today and let the magic of music take you on a journey like never before.

**Collaboration: Critique & Suggestions**

- **Advertiser Agent P/C/A**$_{\text{system prompt}}$
  Recently, your company is writing advertisements for "$test item title" CD, which has the following description: "$test item description". \n\n Your colleague has written a draft advertisement for this CD. The specific content of the draft is as follows: "$draft advertisement".\n\n However, his ability in your professional field may not be as strong as yours, so this draft may not be perfect. Therefore, please provide him with modification suggestions based on your professional expertise. \n\n Important note:\n 1. Your output should follow the format below: "My suggested revisions: [...]"\n 2. Your suggestion should not exceed 40 words, therefore your suggestion should be specific. \n 3. You only need to provide suggestions for modifications, without directly making the changes for him.
- **Advertiser Agent P**$_{\text{response}}$
  - Emphasize the unique qualities of the CD, such as the innovative sound or the thought-provoking lyrics.\n - Highlight the specific artists featured on the CD to attract fans of their music.\n- Use more vivid language to describe the listening experience and the emotions it evokes.\n - Add a call-to-action to create a sense of urgency and encourage immediate purchase.
- **Advertiser Agent C**$_{\text{response}}$
  - Emphasize the unique selling points of the CD, such as the innovative sound production techniques and the distinctiveness of the artists' styles.\n- Add specific examples of the diverse genres represented on the CD to create intrigue and appeal to a wider audience.\n - Highlight the emotional impact of the music and how it can resonate with listeners on a personal level.\n - Consider adding a call to action that encourages readers to take immediate action and purchase the CD.
- **Advertiser Agent A**$_{\text{response}}$
  - Replace "Yes I Am" with a more attention-grabbing title that reflects the CD's unique qualities. \n - Highlight the CD's standout features, such as the emotional depth of the lyrics and the exceptional vocal performances. \n - Emphasize the CD's versatility by mentioning specific occasions where it can be enjoyed, like road trips, cozy nights in, and lively gatherings.

*Continued on next page*

**Refinement**

• **Advertiser Agent N**system prompt

Your colleagues have provided some revision suggestions for your draft based on their expertise.\n \n Specifically, their suggestions are listed as follows:\n "$revise suggestions". \n\n Please take your colleagues' suggestions into account and make revisions to your draft.\n Important note: \n 1. Please generate your revised advertisement directly, without any additional thinking process. \n 2. Your revised advertisement draft should not exceed 200 words. \n 3. If you feel that their opinions hold no value, you can also choose not to heed their advice.\n 4. While considering your colleagues' suggestions, it is important to stay true to yourself and avoid being misled by their mistakes.

• **Advertiser Agent N**response

Immerse yourself in the raw emotion of heartfelt lyrics and the exceptional vocal performances that will resonate deep within your soul. \n With an innovative blend of genres, "Yes I Am" showcases a diverse range of music, from the infectious beats of pop and rock to the soul-stirring melodies of R&B and country. Each track is meticulously crafted, utilizing cutting-edge sound production techniques that will transport you to another dimension.you're embarking on a road trip, enjoying a cozy night in, or hosting a lively gathering, "Yes I Am" is the perfect companion for any occasion. Let the music wash over you, evoking a whirlwind of emotions and connecting you to the very essence of the human experience.\n Featuring an extraordinary lineup of renowned artists, including chart-topping sensations and rising stars, "Yes I Am" is a testament to the power of music. Don't miss your chance to own this extraordinary collection. Take action now and embark on your own "Yes I Am" Get your copy today and let the magic unfold.

**Evaluation**

• **Test User Agent**system prompt

You are a CD enthusiast. Here is your self-introduction, exhibiting your preferences and dislikes: "$user description". Now, you are viewing an advertisement: "$item ads". \n\n Does this advertisement capture your attention? \n\n Please generate your output in the following format: "Choice: [yes / no] \\n Explanation: [Explanation of why you make such choice]". \n\n Important note:\n 1. Please carefully consider each aspect before making your decision. \n 2. If you answer "yes," it means you consider this advertisement to be informative, creative, and aligned with your preferences. Otherwise, you should respond with "no".\n 3. Please make deep analysis.

• **Test User Agent**response to draft ads

No. While the advertisement does highlight a diverse range of genres, including pop and rock, which align with my preferences, it lacks specific details about the artists and tracks featured on the "Yes I Am" CD. Without this information, it is difficult for me to gauge whether the CD will truly captivate my senses and leave me craving for more. Additionally, the advertisement focuses more on the overall experience and production quality rather than the specific qualities of the music itself, which is a key factor for me in deciding whether to purchase a CD. Therefore, I am not sufficiently convinced by this advertisement to consider purchasing the "Yes I Am" CD.

• **Test User Agent**response to refined ads

Yes. This advertisement captures my attention because it aligns with my preferences for CDs that offer a diverse range of genres and exceptional vocal performances. The mention of heartfelt lyrics and the ability to evoke emotions resonates with my preference for CDs with meaningful and introspective lyrics. The promise of an innovative blend of genres, including pop, rock, R&B, and country, appeals to my love for CDs that showcase a variety of musical styles. Additionally, the mention of cutting-edge sound production techniques and the use of renowned artists further piques my interest in experiencing a vibrant and dynamic listening experience. Overall, this advertisement seems informative, creative, and aligned with my preferences, making me inclined to consider purchasing "Yes I AM".

