# OpenReview forum: "AgentCF: Collaborative Learning with Autonomous Language Agents for Recommender Systems"
_ACM.org/TheWebConf/2024/Conference — TheWebConf24 Oral_

### Official Review · Reviewer_sVXi · 2023-11-20

**Novelty:** 6
**Technical Quality:** 5

**Review:**

SUMMARY
In this paper, the authors propose an agent-based framework that exploits Large Language Models to simulate the interaction of a user with a recommender system. In particular, the authors introduce the novel and intriguing idea of seeing also the item as an agent, with its own beliefs and representation. Another interesting idea that is spread with the paper is that gradient optimization can be simulated by means of a multi-turn interaction. The experiments confirmed the effectiveness of the intuitions and the novelty of the approach, while some of the aspects (detailed comments follow) are not completely convincing.

STRONG POINTS
- Novel and timely topic
- Solid methodology, that borrows concepts from agent-based architectures
- Good experimental results

WEAK POINTS
-Sub-optimal choices for the experiments. Limited findings.

DETAILED COMMENTS
- The impact of sampling on the overall results is not clear
- Line 209: Different from traditional recommendation models, the LLM that 208
implements 𝑓LLM (·) will be fixed during the optimization process. - what do you mean by fixed?
- Line 412: At each step of interaction (i.e., optimization), we iterate the 411 selection process (Section 2.2.2) and the collaborative reflection 412 process (Section 2.2.3)  --> do you repeat the process on the same item, or do you continue with a new one?
- Line 734: typo - copmarison
- The main concern I have with the training process is the collaborative reflection part. It should be better explained how and when the process is started. In particular, it is not clear to me in which moment of the training the information coming from other users and prompted to change the representation of the user itself and the items. I guess that the order has some importance here.

**Questions:**

- Please clarify the choice of the dataset.
- Line 209: what do you mean by "fixed" recommendation model?
- Line 412: At each step of interaction (i.e., optimization), we iterate the 411 selection process (Section 2.2.2) and the collaborative reflection 412 process (Section 2.2.3)  --> do you repeat the process on the same item, or you continue with a new one?
- How does the system perform in a complete cold-start scenario (i.e., no ratings for a particular user)
- Please better explain when and how the collaborative reflection part comes into play. It is not clear to me the exact moment when the information coming from peers is exploited to represent the current model of the user.

**Ethics Review Description:**

No ethical concerns

**Reviewer Confidence:**

3: The reviewer is confident but not certain that the evaluation is correct

**Scope:**

3: The work is somewhat relevant to the Web and to the track, and is of narrow interest to a sub-community

---

### Official Review · Reviewer_SKAF · 2023-11-21

**Novelty:** 5
**Technical Quality:** 5

**Review:**

Pros:

1. The paper explores a significant and timely research area by delving into LLM-based recommendation systems and introducing an interesting concept, the item agent.

2. Providing an anonymous code repository adds value to the paper by enhancing reproducibility.

3. The authors present comprehensive experiments, effectively conveying their findings through well-organized tables and graphs.

Cons:

1. Certain crucial concepts and claims lack sufficient explanation, diminishing the paper's clarity.

2. Although Figure 1 illustrates the workflow of the AgentCF framework, its complexity may confuse readers. The figure doesn't effectively highlight the pivotal role of item agents, and the inclusion of extensive text within the figure is discouraged as it diminishes the advantages typically associated with visual representations.

**Questions:**

1. The authors mention that "item information is relatively stable through time". Based on this, what is the necessity of the item agent than the description or item title?

2. "We find that LLMs tend to over-complain the negative item agent's drawbacks, disregarding the fact that it may also be attractive for other users." How do the authors get this conclusion? Are there any citations or experiments supporting this claim?

3. What do $R_B$, $R_{B+R}$ and $R_{B+H}$ stand for? Why $R_{B+H}$ does not need long-term memory in Eq. (8)?

4. $R_{B+R}$ use more information ($M_u^r$) in Eq. (7) than $R_B$ in Eq. (6). Why $R_{B+R}$ performs worse than $R_B$ in some cases?

**Reviewer Confidence:**

3: The reviewer is confident but not certain that the evaluation is correct

**Scope:**

4: The work is relevant to the Web and to the track, and is of broad interest to the community

---

### Official Review · Reviewer_Fagi · 2023-11-23

**Novelty:** 6
**Technical Quality:** 6

**Review:**

**Summary:**

AgentCF proposes a novel approach for simulating user-item interactions in recommender systems through agent-based collaborative filtering. The paper creatively considers both users and items as agents, developing a collaborative learning approach that optimizes these agents together. The proposed methodology involves prompting user and item agents to interact autonomously, followed by collaborative reflection to adjust any misleading simulations based on real-world interaction records. Extensive experiments validate the effectiveness of the proposed approach. It demonstrates that AgentCF achieves superior or comparable performance to traditional recommendation models when trained on similarly scaled datasets. Notably, the model performs effectively with sparse data scenarios, training on only approximately 0.07% of the complete dataset.

**Pros:**

1.	The paper's approach of treating both users and items as agents in a recommender system is highly original. This dual-agent perspective is a significant departure from traditional recommender systems that typically focus on user preferences without equally emphasizing the role of items as interactive agents.
2.	Incorporating memory modules for both user and item agents is an inventive aspect of the paper. This feature allows for the integration of collaborative signal in the user-item interactions. Utilizing language to explicit model and refine CF signals is a relatively unexplored concept in the field of recommender systems. And the introduction of a collaborative reflection mechanism as a training process for mutual memory update between user and item agents is a novel element. This process allows the both user and item agents to adjust their behavior based on discrepancies between simulated and real-world interactions, adding a layer of sophistication to the model.
3.	Experiments validate that AgentCF is effective in the scenarios of few data available, especially in the sparse dataset.
4.	The paper is well organized, clearly written and easy to understand.

**Cons:**

1.	The contribution is limited by the overly sampling of the datasets (only select 1% of the user in the original dataset). In such small dataset, the experimental results which compared to traditional baselines are unconvinced.
2.	Continously updating and reflecting the memory during simulation leads to a overwhelm of text information in the memory of both users and items. And it is highly potential that catastrophic forgetting occur to fit recent interaction without consideration of early interaction. Well-designed mechanism should be apply to effectively utilize the information contained in the memory.
3.	The paper could delve deeper into the interpretability of the model's decisions and recommendations. Given the complexity of the model's collaborative reflection and autonomous interactions, providing insights into why certain recommendations are made could enhance its applicability and trustworthiness.

**Questions:**

AgentCF demonstrates commendable performance in situations where training data is sparse. Its success is largely due to leveraging a Language Model (LLM) that's pre-trained on extensive corpora, thus efficiently utilizing semantic information in few-shot scenarios. However, there's a concern regarding its enhanced performance with increased data. Given that LLMs, despite their robustness, might struggle with longer prompts, the rising complexity and length of inputs could adversely affect the LLM's reasoning capabilities and context maintenance. This issue could lead to reduced performance gains as the data volume grows, potentially impacting the overall contribution of this work. It would be beneficial if the authors could address this challenge, potentially improving the system's adaptability to diverse data volumes and further solidifying the work's contribution.

**Reviewer Confidence:**

4: The reviewer is certain that the evaluation is correct and very familiar with the relevant literature

**Scope:**

4: The work is relevant to the Web and to the track, and is of broad interest to the community

---

### Official Review · Reviewer_KRAK · 2023-11-24

**Novelty:** 5
**Technical Quality:** 6

**Review:**

This paper proposed an agent-based collaborative filtering approach for simulating user-item interactions. In the proposed method, they leveraged the capabilities of LLMs to optimize the simulated user agents and item agents, so their preferences can be propagated in subsequent interactions. They evaluated their proposed approach on two datasets and compared it with several baseline methods to demonstrate its effectiveness.

**Pros:**
1. The investigated topic of user behavior simulation is interesting and will be of interest to researchers in recommender systems.
2. The proposed model learns the user-item interaction by autonomous interaction between user and item agents and enables optimization of user and item agents to capture two-sided relations.
3. The evaluation and ablation study results show the proposed approach can simulate user-item interactions, achieving good performance on recommendation tasks.

**Suggestions for improvements:**
1. The presented framework of AgentCF in Figure 1, is a bit confusing to me with the current figure caption. The authors might consider adding more detailed explanations for the figure. For example, it is unclear how the figure shows “the simulated preferences of user and item agents aggregate (as indicated by the highlighted content) and can propagate to other agents in subsequent interactions.”
2. For the experiment part, it is better to explicitly specify what questions they want to answer in the whole experiment. It would be easier to understand the rationale behind the experiment.
3. For the baseline methods, it would be better to include more explanation on how the baseline methods were chosen for the experiment.
4. For the ablation study, it seems the results only show AgentCF_B and the associated comparison instead of the other two prompting strategies. It is better to provide some explanations in the paper.
5. For the related work section, it might be better to add some discussion on existing work on user behavior simulation.

**Questions:**

1. For the baseline methods, I wonder how the baseline methods were chosen for the experiment or the rationale behind the choice.
2. For the ablation study, it seems the results only show AgentCF_B and the associated comparison instead of the other two prompting strategies. I wonder about the reason for this selection.

**Reviewer Confidence:**

2: The reviewer is willing to defend the evaluation, but it is likely that the reviewer did not understand parts of the paper

**Scope:**

3: The work is somewhat relevant to the Web and to the track, and is of narrow interest to a sub-community

---

### Decision · Program_Chairs · 2024-01-22

**Decision:**

Accept (Oral)

**Comment:**

The paper introduces AgentCF, an innovative agent-based collaborative filtering approach for simulating user-item interactions in recommender systems. This method creatively leverages the capabilities of Large Language Models (LLMs) to optimize simulated user and item agents, thus enhancing the recommendation process. The proposed approach has been thoroughly evaluated on two datasets and demonstrates its effectiveness through comparison with several baseline methods. The paper's strengths lie in its originality and relevance to current research trends in recommender systems. The dual-agent perspective, treating both users and items as agents, represents a significant departure from traditional models, offering a more comprehensive understanding of user-item dynamics. The incorporation of memory modules for both agents, allowing for the integration of collaborative signals, is an inventive and valuable addition to the field. Furthermore, the experiments effectively validate the model's efficiency, particularly in sparse data scenarios. There are areas that could be improved for enhanced clarity and robustness. The presentation of the AgentCF framework in Figure 1 could be made clearer, with more detailed explanations to aid comprehension. The paper could benefit from more explicit definitions of the experimental objectives and a deeper discussion on the selection of baseline methods for comparison. The ablation study results should include more details about the different prompting strategies used. Additionally, the paper could explore further the interpretability of the model's decisions and recommendations, which would enhance its applicability and trustworthiness.